# Snakebites in Rural Areas of Brazil by Race: Indigenous the Most Exposed Group

**DOI:** 10.3390/ijerph18179365

**Published:** 2021-09-05

**Authors:** Maria Cristina Schneider, Myriam Vuckovic, Lucia Montebello, Caroline Sarpy, Quincy Huang, Deise I. Galan, Kyung-Duk Min, Volney Camara, Ronir Raggio Luiz

**Affiliations:** 1Department of International Health, School of Nursing and Health Studies, Georgetown University, Washington, DC 20057, USA; mv225@georgetown.edu (M.V.); cas480@georgetown.edu (C.S.); qh32@georgetown.edu (Q.H.); dgl32@georgetown.edu (D.I.G.); 2Institute of Collective Health Studies, Federal University of Rio de Janeiro, Rio de Janeiro 21941-598, RJ, Brazil; volney@iesc.ufrj.br (V.C.); ronir@iesc.ufrj.br (R.R.L.); 3Department of Immunization and Transmissible Diseases, Ministry of Health of Brazil, Brasilia 70723-040, DF, Brazil; lucia.montebello@gmail.com; 4Institute of Health and Environment, Graduate School of Public Health, Seoul National University, Seoul 08826, Korea; kdmin11@hotmail.com

**Keywords:** envenomation, snakebite, race, complications, time between accident and health care, Brazil

## Abstract

Animal stings are environmental hazards that threaten millions annually and cause a significant socioeconomic impact. Snakebite envenoming affects 2.7 million people globally every year, mostly the poorest and rural communities, with approximately 27,000 annual cases in Brazil. This study’s objective is to identify the most exposed racial group for snakebites in rural areas of Brazil and analyze possible differences in the outcome of an accident. A retrospective epidemiological study was conducted using a database of rural snakebite cases from Brazil’s Ministry of Health (2017). Descriptive analysis and a regression model were performed to examine the association of bad outcomes after a snakebite with several covariables. While mixed-race individuals presented the highest number of cases (61.79%), indigenous and white populations were the racial groups with the highest and lowest exposure rates (194.3 and 34.1 per 100,000 population, respectively). The fatality rate was 3.5 times higher in the indigenous population compared to the white population. In the multivariable model, the number of hours between the accident and health care received and the case classification suggested an association with a bad outcome. Snakebite is prominent in Brazil, particularly among indigenous groups. Antivenom is available in the Brazilian Health System; however, efforts need to be made for decentralization.

## 1. Introduction

Animal bites and stings are important environmental hazards that threaten millions of people annually and cause a significant socioeconomic impact at the individual and country level. Snakebite envenoming is one of these examples that affects close to 2.7 million people globally every year and has an estimated mortality close to 100,000 deaths [1,2]. Approximately 95% of snakebite envenoming occurs in low- and middle-income countries, and snakebites disproportionately involve the poorest of the poor, mostly in rural areas [3]. Improperly treated, envenomation poses a large threat to public health in terms of suffering, morbidity, mortality, and long-term disability [2].

In response to the large global burden of disease due to snakebite and the global crisis in antivenom production that has left millions of vulnerable people with no or difficult access to the antivenom, the World Health Organization (WHO)’s Strategic and Technical Advisory Group for Neglected Tropical Diseases recommended the reclassification of snakebite envenomation as a high priority Neglected Tropical Disease (NTD) at the WHO’s seventy-first World Health Assembly in 2018 [4]. This classification identifies snakebite envenoming as a substantial obstacle to several of the Sustainable Development Goals (SDGs) by 2030 [5], including but not limited to the SDGs related to poverty, health, and education. Therefore, the WHO requires a more comprehensive overview of the epidemiology of snakebite envenoming to better identify and address populations in need of greater preventative education, treatment, and long-term management services [4]. Furthermore, there is an urgent need to improve antivenom availability, accessibility, and affordability on a global basis to ensure safe and effective pharmacotherapeutic treatments to snakebite victims, one of the four pillars in WHO’s global strategy [4,6]. The lack of long-term investment in the proper management of snakebite envenoming, as well as financial and commercial factors, have contributed to the crisis in antivenom production, with detrimental consequences for individual victims, their families, and communities [7].

In the Americas, snake envenoming is a serious public health threat that has been recognized for many years, as the variety of venomous snakes, as well as environmental and socioeconomic conditions, make the region particularly vulnerable to snakebites [8,9,10,11,12,13,14,15,16]. A previous study on the America region estimated an average annual incidence close to 57,500 snakebites (6.2 per 100,000 population) and 370 reported deaths per year (0.04 per 100,000 population), with the severity of the bites mainly depending on the species and size of the snake and the accessibility to health care, including the availability of antivenoms [12]. Latin America has a long tradition in snake antivenom production; there are public and private laboratories that supply these products to most of the region’s countries [17]. In the last decades, with the increase of the regulatory requirements in immunoglobulin production, some laboratories had to stop operation until they met these requirements. In 2019, a network was created to support the production of antivenom in the region; one of the objectives of the network is to contribute to the laboratories’ ability to meet all regulatory requirements in the production, to increase the availability and access to effective and safe antivenoms throughout Latin America [17].

Recognizing snakebite as an important public health problem, the Ministry of Health of Brazil created the National Program for Snakebite Control in 1986 and extended the program in 1988 to include other venomous animals [18]. Envenoming accidents caused by animals such as snakes, scorpions, spiders, and others are subject to compulsory notification to the Ministry of Health of Brazil Notifiable Diseases Information System, National Health Surveillance Secretariat database (acronym in Portuguese, SINAN) [19].

Brazil suffers from the fourth-highest number of snakebites in the world, particularly within its rural, agricultural populations [1]. Currently, around 27,000 snakebite cases per year are reported countrywide, with the highest number of cases and rates in the Legal Amazon Region [16]. The One Health approach was used in previous studies to understand the complexity of snakebite in Brazil, demonstrating the association with tropical habitat, lower percentage of urbanization, lower Gross Domestic Product (GDP) per capita, forest loss, and with venomous snake richness, suggesting that Brazilians living in remote, tropical areas are at a higher risk to be bitten by a snake [16].

Antivenom and health care are provided free of charge by the Brazilian health system but access to the lifesaving serum is not always available at the local level and poses a particular problem in remote areas [9,16]. Brazil has a decentralized universal public health care system—the Unified Health System (also known by its Portuguese acronym, SUS)—used by nearly 65 percent of the population [20]. The health care system offers all Brazilians free access to appointments, hospitalization, and a wide range of medicines, in addition to vaccination campaigns, and prevention and health surveillance actions. Nevertheless, debates about inequalities in the distribution of health services and on the differences in health status among different population groups have figured prominently for decades [20].

Recent studies and analyses of health indicators show that despite decentralization efforts from the federal government, Brazil’s indigenous population and Brazilians of African descent still face greater health inequities than their white counterparts [21,22,23,24,25]. Because indigenous groups live largely in rural, and oftentimes remote areas and work in high-risk environments such as subsistence farming, hunting, and gathering, they are frequently among the reported victims of snakebite and might be at greater risk of accidents with these animals than other population groups [26,27]. We believe that indigenous groups are more exposed to snakebites than other population groups; however, to date, this burden has not been measured adequately, and very few studies have taken race and ethnicity into account in their analysis of snakebite in Brazil [28,29].

Brazil is a vast country and the differences among its regions are large, not only ecologically but also culturally and ethnically. Prior studies have explored the demographics of snakebite victims in Brazil [9,18,30,31]; however, none has so far focused on the association with race and ethnicity. The Brazilian surveillance system SINAN includes ‘race’ as one of their variables in the notification form filled out when a person receives health care after a snakebite in the SUS. The variable ‘race’ collected in SINAN is self-declared, just as in Brazil’s census, which is administered every ten years by the Brazilian Institute of Geography and Statistics (acronym in Portuguese, IBGE) [32].

The objective of this study is to identify which racial groups in rural areas of Brazil are more exposed to snakebites and to compare the most exposed group with the less exposed group by region. We also analyze if there are possible differences in the outcome of the snakebite, considering race and other variables related to the accident.

## 2. Materials and Methods

### 2.1. Study Area

Brazil is the largest country in Latin America with a geographic area of 8,515,767 km^2^ and a population of 190,755,799, according to the latest census conducted in 2010 by the Brazilian Institute of Geography and Statistics [32], distributed in five regions, 26 states, and the Federal District. In 2010, the Southeast Region concentrated 42.1% of the total population. The largest population groups in Brazil by race are the white population (47.7%), followed by the mixed-race population (43.1%). Black and Asian Brazilians constitute 7.5% and 1% of the population, respectively. According to the last census, the indigenous population included 896,917 individuals in the entire country, making up only 0.4 percent of Brazil’s population [32]. While Brazil’s indigenous population is relatively small, it is ethnically diverse and includes 300 indigenous ethnic groups with speakers of over 200 distinct languages [22].

The majority of Brazil’s population lives in urban areas, with only 15.6% residing in rural areas. There are, however, important differences among the regions, with close to 27% of the population living in rural areas in the Northeast and North Regions compared to only 7.1% in the Southeast (Appendix A). Most indigenous people in Brazil live in over 600 federally recognized reserves, 98% of which are in the Amazon Region, but a substantial number of indigenous individuals (42.3%) live outside of recognized reserves, and most of the indigenous population lives in rural areas (73.8%) [22].

Around 3% of the total population are agriculture workers [32]. Considerable differences in literacy exist, ranging from 95.9% of the population in the South to 83.8% in the Northeast Region, with a country average of 92.0% [33]. Large differences are also apparent with regards to the GDP per capita, ranging from BRL 34,790 in the Southeast to BRL 12,955 in the Northeast Region (Appendix A) [34].

The major habitat type in Brazil is tropical and subtropical moist broadleaf forests (TSMBF) [35]. This habitat type includes the Legal Amazon Region and some areas of the Atlantic Forest in the coastal area, where close to 70% of the snakebites occurred [16]. According to the Ministry of Health of Brazil, the venomous genera with public health interest in Brazil are *Bothrops*, *Crotalus*, *Micrurus,* and *Lachesis* [36,37]. *Bothrops* is present in most of the country; *Crotalus* is least present in the Amazon Region; *Micrurus* is least present in the Northeast Region, and *Lachesis* is not present in the South Region. According to previous studies, approximately 70% of snakebite envenoming is caused by *Bothrops* snakes [9,29,30,37].

The Brazilian Unified Health System runs from primary care to high complexity health services countrywide [24]. In case of an accident with a venomous animal, health care is delivered by SUS and the antivenom and other treatments are provided free of charge for the population [37]. Around 1600 hospitals nationwide, located in all 26 states and the Federal District, provide five types of antivenom for different types of venom, including five types of snake antivenoms: antibothropic pentavalent, antibothropic pentavalent and antilachetic, antibothropic pentavalent and anticrotalic, anticrotalic, and antielapidic (Appendix A) [36]. However, due to a lack of health professionals and infrastructure in remote areas (e.g., electricity to ensure an adequate cold chain), particularly in the Amazon Region, access to lifesaving antivenom is limited, leading to a delay of patient care or no medical treatment at all [18,37].

Indigenous health is part of the SUS through the Special Secretariat for Indigenous Health (SESAI), with the participation of the state and municipal health authorities [35]. The Special Indigenous Health District (in Portuguese DSEI: Distritos Sanitários Especiais Indígenas) is the decentralized management unit of the Indigenous Health Care Subsystem (SasiSUS). In Brazil, there are 34 DSEI, strategically divided by territorial criteria, based on the geographical location of indigenous communities. The DSEI do not align with the borders of the states but cut across state lines. This structure for indigenous health care includes around 360 basic indigenous health units in the reserves, centers, and the Indigenous Health House at the national level (in Portuguese, CASAI) [38,39]. This structure manages health care of different complexities and includes provisions for transportation (usually by plane) to other places when needed. In the case of an accident with a serpent in the DSEI’s jurisdiction, transportation to a hospital with the capacity to administer the antivenom is usually required.

### 2.2. Study Design and Source of Information

We carried out a retrospective epidemiological study using the SINAN database by individual cases of accidents with venomous animals from the Ministry of Health of Brazil. The database was requested via the Citizen Information System [40]. According to the information access law (in Portuguese “Lei de Acesso a Informacao” number 12,527 from 18 November 2011), government information can be requested (https://www.gov.br/acessoainformacao/pt-br) (accessed on 19 March 2019); in this case, the protocol number was 25820.001698/2019-76, of 2019. The notification form collects information about the patient’s demographic characteristics, epidemiological data about the accident, case classification, the time between the accident and receiving health care, whether the patient received antivenom and the case evolution. All cases are coded and do not reveal any information related to the identification of the person for ethical reasons.

The data include information from several types of accidents with venomous animals such as snakes, spiders, and scorpions. For this study, only snake accidents were selected, and only cases reported in the most recent year in the database (2017). From the total number of cases of snakebite accidents in Brazil in 2017, we selected all rural cases, and among them the ones with information about race. Data on the population by race and by state were obtained from the 2010 national census [32].

The Ministry of Health provides guidelines to health care professionals on how to complete the notification forms, and on the data to be gathered. As cited, the variable ‘race’ collected in SINAN and in the IBGE census is self-declared, and both institutions combine the categories of ‘race’ and ‘ethnicity’ into one.

Our study population is all snakebite cases in Brazil reported to SINAN in rural areas during the year 2017, which have information about the racial identification of the victim [19].

### 2.3. Parts of the Study

#### 2.3.1. Part 1: Descriptive Analysis

In the descriptive analysis of the epidemiological situation, cases of snakebite and rates per 100,000 population were described by race, by region, and by state in rural areas of Brazil, and the rate ratio by race compared with less exposed racial groups was calculated. To avoid unstable rates with small populations (under 1000 inhabitants in the specific race group), rates were not estimated for small populations and considered ‘not applicable’ (NA) in the tables by state.

Another part of the descriptive analysis presented covariables related to demographics and the accident, including the bite site on the body, the type of snake related to the accident reported, the number of hours between the accident and health care received (the cut-off was pre-defined by the Ministry of Health in the database), case classification, and if antivenom was administered. The definition of case classification according to the Ministry of Health of Brazil [36] is by the genera of the venomous snake and includes recommendations on what type of antivenom and the number of ampoules to be used for treatment (Appendix A). This information is available in the notification form that the physician fills out when an accident happens and in the Ministry of Health guidelines [36]. In the last part of the descriptive analysis, possible differences related to the outcome of the accident were presented, including if there were local complications (such as secondary infections, extended necrosis, and amputation), systemic manifestations (such as acute renal failure, sepsis, and shock), and the evolution of the case (cure, death by snakebite, death by other causes or ignored). The total fatality rate by snakebite in rural areas of Brazil was estimated and calculated for the most exposed and less exposed racial groups.

Descriptive analysis was conducted using Stata IC/16 and some figures in Microsoft Excel for Microsoft 365 MSO.

For better visualization of snakebite case distribution by state, GIS maps were prepared for the total number of cases, and for cases in the most and less exposed groups, using population data from IBGE as background. The maps were created using ArcGIS versus 10.8, utilizing natural breaks for the population and case ranges.

#### 2.3.2. Part 2: Statistical Analysis

As the number of deaths was much smaller than the number of cases reported, a new variable was created considering the outcome as a combination of three possibilities: the evolution (0 = cure, no death by snakebite; 1 = death by snakebite), and/or systemic complication (0 = no; 1 = yes), and/or local complication (0 = no; 1 = yes).

A binomial logistic regression model was used to estimate the association between the outcome of a snakebite accident (0 = good; 1 = bad) and the covariables race, sex, age, bite site on the body, number of hours between the accident and health care received, case classification, and if antivenom was administered. One person could have one or more of the situations considered as a bad outcome (possible complications and death). Only observations that have all the independent variables completed in the database were included.

In this part of the statistical analysis, the mixed and black racial groups were included as a combined group under the designation “black” to simplify the results of the logistic regression. Due to the small number of cases in the Asian racial group (127 cases, 0.8%), this group was excluded from the analysis as it was not relevant to the study, given that they were neither the most nor less exposed racial group. We created categories for the variable age, and we combined some of the existing categories of the variables related to bite site on the body and the number of hours between the accident and health care received.

A univariate analysis was conducted followed by a multivariable analysis, and estimation of the odds ratio (OR) with a 95% confidence interval (CI). The statistical analysis was performed using SPSS 24.

## 3. Results

### 3.1. Part 1: Descriptive Analysis

In 2017, the total number of snakebites in Brazil reported to SINAN was 28,716, with 16,805 of them (58.52%) recorded in rural areas (Appendix A). The One Health approach showing the major habitat types, presence of venomous snakes by genera, GDP per capita, and the number of rural cases of snakebites by region is presented in Figure 1. Analyzing this database, 75.34% of the cases were caused by genus *Bothrops*, and these percentages are similar in all regions. Accidents with *Micrurus* constituted 61.86% in the Northeast Region, and accidents with *Crotalus* were more frequent in the Northeast and Southeast Regions (39.01% and 34.46%, respectively). Accident by *Lachesis* were predominant in the North Region (90.01%) (Appendix A).

The total number of rural observations of snakebites with information on the race of the victim was 15,898, which constitutes our study population. The North Region, which is part of the Amazon Legal Region, has the highest number of snakebite cases in Brazil (Figure 2). The distribution of snakebite cases by state demonstrates that Para state in the North Region has the highest number of cases (3420 cases), followed by Minas Gerais (1710) in the Southeast Region, and Bahia (1534 cases) in the Northeast Region (Appendix A).

The racial group with the most cases was the mixed racial group with 9823 (61.79%), and the racial group with the lowest number of cases was the Asian group with 127 recorded snakebites (0.80%) (Appendix A). However, when estimated as rates per 100,000 population, the racial group most exposed to snakebite was the indigenous population, with a rate of 194.3 per 100,000 population (977 cases), and the less exposed racial group was the white population with 34.1 per 100,000 population (3701 cases) (Table 1). The rate ratio comparing the most exposed and the less exposed racial group was 5.7, suggesting that the probability of an indigenous individual being bitten by a snake is approximately six times higher than that of a white person in Brazil. In the Northeast Region this probability is even higher (8.9 times), whereas it is lower in the Southeast (1.8 times) and South (2.5 times) Regions.

The type of snake related to the accident by racial group is presented in Table 2, calling to attention that the genus *Lachesis* presented higher percentages in the indigenous populations (5.22%) and *Crotalus* smaller (5.42%), compared with the total population (2.16% and 7.8%, respectively), and with the white populations (0.46% and 9.59%, respectively).

Snakebites affected mostly males (76.4%), and the average age across all populations was 36 years old, with the white population tending to be older than the indigenous population. Half (55.2%) of all bites occurred in lower limbs, particularly in the feet (Figure 3); for the indigenous population, this percentage was 66.1% compared to 49.4% for the white population. Over 75.2% of the white population received care between 0 and 3 h after being bitten, compared with only 37.4% of indigenous individuals who received care during this timeframe.

Related to the case classification, 52.8% of cases were considered mild, 37.6% moderate, and 7.7% severe, without significant differences among the racial groups (Table 3). Among the snakebite cases reported in the SINAN database, 79.7% of the cases received antivenom, with the indigenous population receiving slightly more antivenom than the general population (Table 3).

Our analysis of clinical outcomes found that local complications were reported in 569 cases (3.9%), with indigenous groups presenting a higher percentage (5.9%) of complications. The most frequent local complication reported was secondary infection (73.5%) (Table 4). The number of cases with systemic complications was 173 (1.2%), showing almost no differences among the racial groups; the systemic complication most frequently cited was acute renal failure (*n* = 137, 79.2%) (Table 4).

According to the SINAN database, among the 15,898 cases reported in rural areas in 2017 with information on race, a total of 14,260 cases had information about the case evolution. Of those, a total of 52 snakebite victims died because of their accident occurring in rural areas of Brazil in 2017 (three more cases were reported without race information), indicating a fatality rate of 0.4% in rural areas. The indigenous and white populations recorded the same number of deaths (6 cases in each racial group); however, the fatality rates among the two racial groups were different: an estimated 0.7% for the indigenous and 0.2% for the white group—3.5 times higher in the indigenous group (Table 4).

### 3.2. Part 2: Statistical Analysis

The full outcome of the binomial logistic regression model with all variables estimates is included in Appendix A. In the statistical analysis (*N* = 13,710), the univariate analysis of the indigenous race group suggested an association with a bad outcome of the snakebite accident (OR = 1.63; CI_95%_ = 1.19–2.24); also, an association was suggested with several of the covariables (Table 5). However, in the multivariable model, only the number of hours between the accident and health care received and the case classification suggested association (Table 4). Increasing the number of hours between the accident and health care increased the risk of a possible bad outcome: three to six hours (OR = 1.38; CI_95%_ = 1.12–1.70), six to 24 h (OR = 1.48; CI_95%_ = 1.18–1.86), and more than 24 h (OR = 2.99; CI_95%_ = 2.21–4.06). The classification of the accident as moderate (OR = 3.29; CI_95%_ = 2.65–4.01) or severe (OR = 11.8; CI_95%_ = 9.29–15.97) was associated with a bad outcome, whereas the variable race lost its significance in the multivariable model.

The risk of an accident with a snake resulting in a bad outcome is 4.7% (a total of 644 cases with bad outcome). This percentage increases with the number of hours between the snakebite and the provision of health care, and in severe cases this percentage increased considerably (from 15.2% with health care received until three hours to 35.7% when more than 24 h had passed). The distribution of a bad outcome in the case of snakebite envenoming among the regions suggested that the Central-West, North, and the Northeast Regions presented higher percentages of bad outcomes (5.8%, 5.7%, and 5.7%, respectively) (Table 6). In two states in the North Region, the percentage of bad outcomes related to the snakebites was higher than 10% of the total accidents reported (Table 6).

## 4. Discussion

Snakebite envenoming is a major public health problem in Brazil, affecting close to 30,000 people each year. To save lives, a large number of ampoules of antivenom needs to be distributed regularly across the entire country. The Ministry of Health in Brazil has been distributing around 230,000 ampoules of antivenom to approximately 2000 hospitals free of charge annually [16], which is likely one of the reasons for the low mortality rates associated with snakebites in Brazil found in this and prior studies [9,16,28,29,30,41]. Brazil’s case-fatality rate for snakebites of around 0.4%, found in our study and the previous study by Bochner [9], is particularly low if compared with studies in other countries, such as the reported case-fatality rate of 3% in a recent district-level study in Cameroon [42], and an estimated case-fatality rate of 3.2% for in-hospital cases of snakebite envenoming in India [43]. According to Chippaux [44], the model that exists in Brazil regarding antivenom distribution, as well as existing infrastructure that ensures treatment for many people, is a good model for other low- and middle-income countries, including those in Africa and Asia.

The most exposed racial group in Brazil are indigenous people, who have a risk of being bitten close to six times higher than the less exposed white population group, and a fatality rate of more than three times that of the white population (0.7% versus 0.2%). The actual risk is probably higher, given that it is likely that a significant percentage of indigenous snakebite victims do not seek health care immediately after the accident. Indigenous peoples hold their own cosmologies, which also guide their interpretations for illness processes and cure options [45]. In the Amazon Region, traditional medical practices such as herbalism and shamanism remain widespread, and the limited access to health care and deeply embedded cultural beliefs mean that rural Amazon communities are frequently using traditional medicine (e.g., plants) to treat ailments and diseases [46], including as the first treatment for snakebite, which may aggravate the conditions at the bite site [18]. A 2017 study by da Silva Souza et al. [29] estimated that the underreporting of deaths due to snakebite envenomation in Amazonia is at nearly 30%, due to lack of follow-up with the patient and underlying health conditions (so that snakebite is the secondary cause of death). The referred study also emphasizes that indigenous populations and remote riverside communities seem to be at higher risk due to agricultural activity and that low accessibility of antivenom due to a greater distance from health care centers contributes to higher case-fatality rates [29].

In Brazil, different medical traditions coexist, including Western biomedicine, popular and traditional medicine involving healers and herbalists, as well as shamans [47]. These different systems of medicine are not mutually exclusive and rural patterns of resort include self-care, visits to traditional healers and the use of indigenous medicines, and utilizing modern medicine as a last resort or in combination with indigenous medicine. The use of traditional herbs and herbal extracts to treat *Bothrops* envenoming has been observed in riverside communities in the state of Para, which is the state most affected by snakebites in Brazil [47]. Other studies in Latin America demonstrated that as much as 60% of snakebite cases were treated by traditional healers [8]. Indigenous and rural populations are also known to use tourniquets, chemicals such as alcohol (applied or ingested), as well as puncture or suction of the bite site to remove the venom or reduce its effects [26].

Access to health care resources in the Amazon Region is still a challenge in Brazil and its neighboring countries [21,25,26,46,48], and a more inclusive health care system is needed, which focuses on rural, low-income populations, including indigenous people and mixed-race riverside communities. This is particularly urgent, as close to 40% of Brazil’s indigenous population lives outside of the designated indigenous reserves and is therefore not covered by the services provided through the Special Indigenous Health Districts. Even though over the past three decades, Brazil’s Unified Health System has undertaken efforts to guarantee more equitable access to health services for the indigenous population through the establishment of the Indigenous Health Subsystem, indigenous Brazilians still suffer from the worst health status of any population group [21,22]. Health care should be culturally sensitive, targeted, and tailored to the needs of the indigenous and other low-income populations exposed to snakebites, particularly in remote areas of the North Region. Health education programs should follow the principle of interculturality, incorporating local cultural knowledge to prevent accidents, and to inform the work of primary health agents in innovative activities at the local level, in support of the national program providing the antivenom.

Transdisciplinary studies, using the One Health approach, are recommended to better understand snakebite in the context of communities living in close contact with nature, particularly Brazil’s indigenous peoples. Understanding indigenous perceptions of the causes and consequences of snakebites and the barriers they face in accessing emergency medical care will help to improve the prevention of accidents and reduce the burden of disease caused by snakebites in this population [16,49].

Although in most states, the indigenous population of Brazil is small compared to other racial and ethnic groups, they constitute a sizable population in the states that belong to the Amazon Region. There are officially 34 Special Indigenous Health Districts, and it is crucial that these units of the Indigenous Health Subsystem always have the antivenom available, that health care personnel are trained to administer it, and have the local conditions in place to respond to health emergencies.

However, since rural, multi-ethnic riverside populations in the Amazon are also at higher risk for snakebite [16,26], they too require special attention. Due to the large geographical distances, absence of roads, lack of regular transportation, and remoteness of these communities, provision of adequate primary health care services has been challenging. Over the past decades, the Brazilian health authorities have implemented fluvial mobile units as an alternative health care model to increase access and health care coverage in the Amazon basin [25]. Although access to routine services has improved through the monthly visits of the Fluvial Family Health Teams, due to the intermittent nature of the visits, the fluvial mobile units cannot be relied upon to provide the needed emergency services in the case of severe snakebite accidents. At the same time, the only health professionals permanently stationed in these remote areas are community health workers and sometimes nurses and nurse assistants. These health care workers are often based at smaller health centers that lack access to stable electricity and therefore storage facilities for antivenoms and other drugs, which require refrigeration [25,50]. This means that snakebite victims frequently need to travel for 12 h or more—reportedly up to 30 h—via boat to the nearest town or city to receive the lifesaving antivenom from a larger health care facility, which often leads to serious complications and negative health outcomes [50]. Our results presenting the genera of the snakes related to the accidents demonstrated that among the indigenous population, the percentage of cases caused by *Lachesis* was more than double compared with the total population; accidents with *Lachesis* are always classified as moderate to severe [36]. The indigenous reserves are located in areas where *Lachesis* are more present geographically, mostly in the Amazonian Region, confirming the urgent need to have the antivenom available on the reserves.

In our study, the region that presented the highest rate ratio between the two race groups (indigenous and white) was the Northeast Region, which has the lowest socioeconomic indicators and the largest rural population in Brazil. Most studies on indigenous populations in Brazil have been carried out in the Amazon Region [45], and more research should be conducted to understand if the indigenous population living in low-socioeconomic areas outside of the Amazon Basin is even more vulnerable to snakebites and other health issues. In the states of the Southeast and South of Brazil, which have better socioeconomic indicators, the differences among the race groups are smaller but still present in most of the states.

The demographics related to sex and age described in this study are similar to those identified in previous studies [9,14,30] and by the Ministry of Health of Brazil [37]: snakebite cases are most common among males (76.4%) in the productive age group (average age of 36 years).

In the study population, the most frequent location of the bites were the lower limbs, particularly the feet, as confirmed in previous studies in Brazil [9,37], whereas studies in other parts of the world also show hands as frequently affected parts of the body [51,52]. This is likely due to the frequent use of sandals in Brazil’s tropical climate, as well as to the custom of many indigenous Brazilians to walk barefoot in the Amazon rainforest—a behavior which is reflected in the higher percentage of bites in the feet among this population group (66.1% versus 49.4%). This finding suggests that for rural communities in countries with tropical climates, more effective primary prevention strategies are needed, such as wearing practical, and at the same time, culturally appropriate protective gear outdoors. Like previous studies, our study confirmed that the majority of accidents with venomous snakes in Brazil are related to the genus *Bothrops* [9,36], and that this is similar in the different racial groups.

The average percentage of cases that received health care within three hours after the accident was 61.2%, which is a positive finding considering the country’s large territory, as well the mobility challenges due to its geography, mainly in the Amazon Region. The difference between the most and less exposed racial groups is great, however, with 75.2% of the white population receiving care between 0 and 3 h after being bitten, compared to only 37.4% of indigenous individuals receiving care during this time. This suggests a delay in medical care due to difficulty in accessing health care facilities with antivenom and other health services for indigenous individuals.

According to Chippaux, “mortality results from both the toxicity of the venom, associated with the amount of inoculated venom, and the precocity and effectiveness of the treatments” (p. 9, [30]). In our study, an increase in time to accessing health care was associated with bad outcomes for accidents with venomous snakes, including severity and mortality. This finding, which aligns with previous studies, is particularly relevant in the North Region of Brazil, where higher numbers of cases occur and venomous snakes such as *Bothrops atrox* and *Lachesis* are common [16,26,30,53].

As expected, in the final statistical analysis, the variables related to the severity of the case classification presented a higher association with bad outcomes. In our study, around half of the cases that arrived in the health system were classified as mild, a percentage similar to the one reported by previous studies and the Ministry of Health [30,37]. However, even though the Ministry of Health guidelines define the criteria for the case classifications related to severity, it has been suggested that in practice, the severity classification is often empirically established, and we propose that more studies should be conducted at the hospital level [25].

In Brazil’s Legal Amazon Region, which includes all seven states of the North Region as well as parts of the Central-West and Northeast Regions, where access to health care is limited and difficult, and the Northeast Region, which has a comparatively low GDP per capita, the percentage of bad outcomes is higher, suggesting that training local health care professionals in the management of envenomation and reviewing the location and number of hospitals and health care centers with the antivenom could substantially improve case outcomes.

It is important to remember that the current antivenoms require conservation in adequate facilities, which are not always available in remote settings. In addition, training of health personnel in antivenom administration, side effect management, and case monitoring is required.

One of the limitations of this study is the use of secondary data from the Ministry of Health. Data may be underreported, such as the number of cases that received health care following a snakebite. Furthermore, a percentage of cases likely did not seek health care, particularly among the indigenous population living in remote areas. The risk of indigenous Brazilians being bitten by a snake and dying from the accident is likely higher than presented in this study and requires further research to measure with greater accuracy the burden of snakebites in this racial group. Additional research should be conducted to examine the likely sub notifications in the number of deaths. However, the SINAN database was also used as a source of information for previous studies on snakebite envenomation [12,16,30,31] and provides the best secondary data currently available. Since the government requests compulsory notification of all snakebite cases and these data are used by decision-makers at the national level to determine the distribution of antivenom to the states, it is the most reliable and complete data source in the country.

Another limitation is the definition of rural areas used by IBGE, which is not very precise but still useful for the purpose of this study, as it provides insight into a particularly vulnerable and often marginalized population, which is considered most at risk for snakebites [2]. The self-declared category of race is another limitation, but since the focus of the analysis was on the comparison between white and indigenous races, the possible bias of classification is lower than if the study had focused on the more ambiguous mixed-race category.

The decentralized distribution of the antivenom to more remote places will only be possible if there is a higher number of ampoules available for purchase by the government. Currently, the amount available for government purchase in Brazil is already reduced due to restrictions in the productive capacity of the laboratories, with only four public laboratories capable of producing the antivenom [16], and, as of the publication of this paper, only one public laboratory actually producing the antivenom in Brazil. The ongoing COVID-19 pandemic has led these laboratories to focus their efforts on COVID-19-related research, likely reducing the amount of antivenom produced. This could affect not only the amount of antivenom available for snakebites but also for spider and scorpion envenomation, which affects more than 120,000 Brazilians each year, as well as fatal infectious diseases such as rabies, which also require serum for post-exposure prophylaxis. A survey with the manufacturing institutions was developed to appraise the antivenom production in public laboratories in Latin America during COVID-19 [6]. Among the different aspects analyzed was the demand from the national public health institutions that remained stable during the pandemic; however, five of the twelve laboratories included in the survey did not manufacture during the first half of 2020. “The reasons for halting the production in these cases were the need to carry out improvements in infrastructure for fulfilling the requirements of Good Manufacturing Practices (GMPs) in four cases and the restrictions to do face-to-face work in one institution” (p. 2, [6]).

In recent years, Amazon deforestation, as well as more comprehensive reporting of snakebite envenomation has resulted in an increased incidence rate [30]; in previous studies, the association of snakebite and tree loss was demonstrated, supporting this argument and reinforcing the need to provide access to the antivenom in remote areas of the Amazon Region [16].

According to Gutierrez [1] and adopted in the WHO strategy [2], confronting snakebite envenoming at a global level demands the implementation of an integrated intervention strategy, which includes the research community, antivenom manufacturers, regulatory agencies, national and regional health authorities, professional health organizations, international funding agencies, advocacy groups, and civil society institutions. In Brazil, important steps have been taken in this direction since the medical scientist Vital Brazil started the production of antivenom in the early 20th century, including numerous scientific studies on accidents with venomous animals, and the creation of the National Program for Snakebite Control in the 1980s with the countrywide distribution of the antivenom free of charge [9,16]. However, the global manufacturing crisis of antivenom and the reduction of the number of laboratories producing antivenom in Brazil pose a great challenge to the equitable distribution of ampoules across the country, and particularly in remote areas of the Amazon Basin.

## 5. Conclusions

Indigenous people are the most exposed racial group to snakebites in rural areas of Brazil and present higher fatality rates as compared to the white population, which is the racial group less exposed to snakebites. Even though the antivenom is available free of charge in the Brazilian Health system in all states, in 2017 there were still 55 (52 with race information) deaths recorded in rural areas among the roughly 16,000 cases of snakebites officially reported. The fatality rate among the two racial groups compared in this study was around three times higher for the indigenous group (0.7% versus 0.2%) than for the white group. However, in the final multivariate model, the only covariables associated with bad outcomes after a snakebite were the number of hours between the accident and health care received and the severity of the case. When controlling for all selected variables, race was not associated. If the indigenous population would be able to receive the antivenom in a shorter amount of time, the differences in fatality rates among the two race groups and number of bad outcomes would likely be smaller, suggesting that an effort to decentralize the antivenom to indigenous reserves is an important step to save lives due to snakebites in remote areas. Efforts should also be made to make antivenom more accessible, mainly to the low-income and rural population in the Amazon Region, as transportation in this region is often slow due to the geographic conditions, increasing the number of hours required to arrive to a hospital with antivenom. However, despite the country’s effort in purchasing and distributing snakebite antivenom for decades, in order to increase this life-saving product decentralization, it needs to be available for purchase. This study clearly shows that snakebite envenoming continues to be an important public health problem in rural areas of Brazil, primarily in indigenous communities, which requires special attention.

## Figures and Tables

**Figure 1 ijerph-18-09365-f001:**
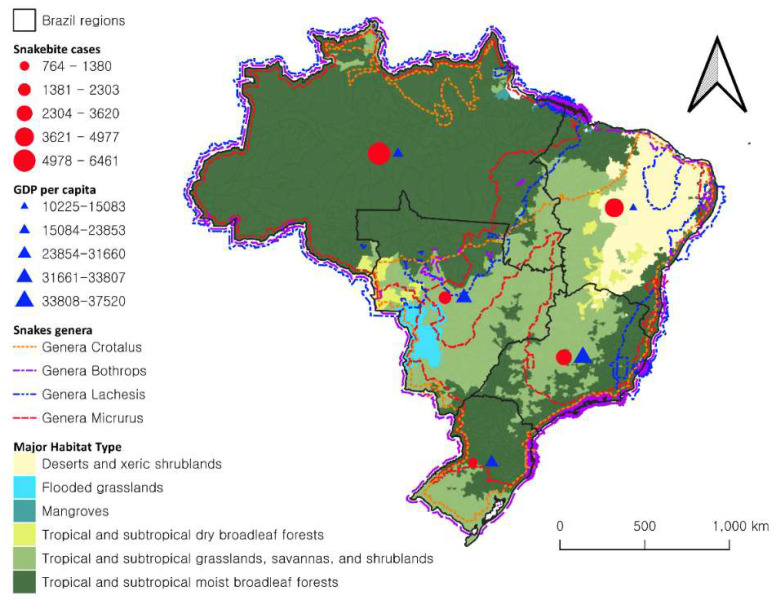
One Health approach to snakebite in Brazil; number of rural cases of snakebite and GDP per capita (Reais) by region, over major habitat type and presence of venomous snakes by genera, Brazil, 2017.

**Figure 2 ijerph-18-09365-f002:**
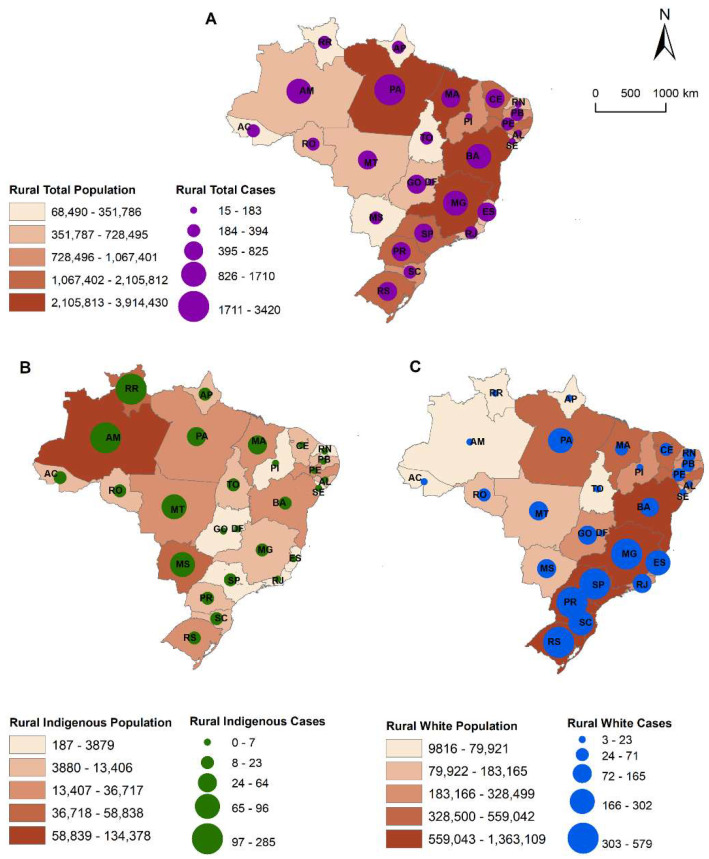
Number of total cases of snakebite (**A**), most exposed race group (indigenous) (**B**) and the less exposed (white) (**C**), rural population, by state, Brazil, 2017.

**Figure 3 ijerph-18-09365-f003:**
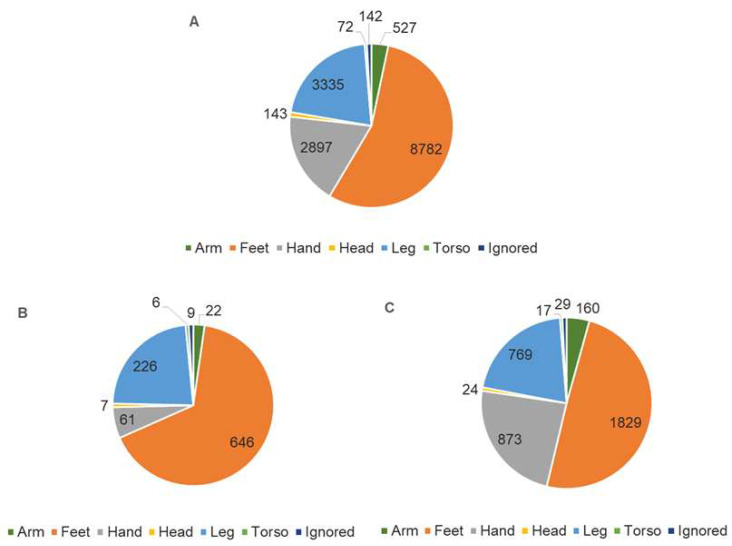
Number of snakebites by site on the body; total (**A**), the most exposed race group (indigenous) (**B**), and the less exposed (white) (**C**), rural population, Brazil, 2017.

**Table 1 ijerph-18-09365-t001:** Snakebite rate per 100,000 population; rate ratio of more exposed race group by less exposed race group per state and per region, Brazil, 2017.

States	Rate by 100,000 Pop						Rate Ratio			
	White	Black	Asian	Mixed	Indigenous	Total	B/W	A/W	M/W	I/W
Rondônia	47.67	59.40	0.00	82.48	142.72	69.45	1.25	0.00	1.73	2.99
Acre	47.93	69.56	32.06	162.28	135.07	131.16	1.45	0.67	3.39	2.82
Amazonas	28.78	87.97	144.40	157.92	212.09	150.72	3.06	5.02	5.49	7.37
Roraima	56.16	186.87	NA	92.60	589.02	286.88	3.33	NA	1.65	10.49
Pará	49.64	162.17	148.64	162.07	164.86	143.13	3.27	2.99	3.26	3.32
Amapá	203.75	422.07	NA	336.28	264.55	316.83	2.07	NA	1.65	1.30
Tocantins	38.62	54.54	184.84	120.81	200.88	103.29	1.41	4.79	3.13	5.20
North	48.24	135.87	122.39	153.84	264.37	140.29	2.82	2.54	3.19	5.48
Maranhão	8.59	33.21	38.59	37.41	225.66	33.98	3.86	4.49	4.35	26.27
Piauí	6.33	8.05	0.00	20.07	NA	15.74	1.27	0.00	3.17	NA
Ceará	12.70	43.30	13.43	30.52	14.87	26.12	3.41	1.06	2.40	1.17
Rio G. Norte	14.14	8.92	0.00	34.27	NA	26.03	0.63	0.00	2.42	NA
Paraíba	10.35	40.30	8.74	32.24	10.28	24.36	3.89	0.84	3.11	0.99
Pernambuco	7.64	20.79	57.88	29.64	25.45	22.59	2.72	7.58	3.88	3.33
Alagoas	2.23	16.71	0.00	21.23	12.08	15.44	7.50	0.00	9.52	5.42
Sergipe	2.31	14.42	0.00	12.98	NA	10.41	6.24	0.00	5.62	NA
Bahia	16.51	54.92	25.26	43.07	114.65	39.19	3.33	1.53	2.61	6.94
Northeast	10.73	38.59	20.56	33.30	95.58	28.50	3.60	1.92	3.10	8.91
Minas Gerais	49.65	79.04	108.57	63.25	70.99	59.33	1.59	2.19	1.27	1.43
Espírito Santo	106.74	103.94	209.21	68.90	35.57	90.15	0.97	1.96	0.65	0.33
Rio de Janeiro	44.52	44.12	25.03	32.06	NA	39.57	0.99	0.56	0.72	NA
São Paulo	43.41	78.64	20.96	39.60	206.24	43.83	1.81	0.48	0.91	4.75
Southeast	52.61	75.47	69.97	56.23	96.27	56.08	1.43	1.33	1.07	1.83
Paraná	38.92	45.82	16.95	16.50	67.13	33.23	1.18	0.44	0.42	1.73
Santa Catarina	28.67	0.00	79.62	10.79	107.64	26.69	0.00	2.78	0.38	3.75
Rio G. Sul	36.24	21.34	13.53	22.31	93.90	35.01	0.59	0.37	0.62	2.59
South	35.15	25.31	26.14	16.79	87.94	32.33	0.72	0.74	0.48	2.50
Mato G. Sul	70.15	64.16	45.89	77.40	141.07	84.71	0.91	0.65	1.10	2.01
Mato Grosso	90.08	134.52	118.46	108.57	261.46	114.25	1.49	1.31	1.21	2.90
Goiás	47.06	97.13	78.39	107.47	NA	83.18	2.06	1.67	2.28	NA
D. Federal	20.52	14.37	0.00	16.05	NA	17.06	0.70	0.00	0.78	NA
Central-West	64.90	99.40	76.18	96.64	185.96	90.72	1.53	1.17	1.49	2.87
Brazil	34.14	60.84	45.21	60.94	194.32	53.30	1.78	1.32	1.78	5.69

Legend: NA: Not applicable in the states with less than 1000 population in these race groups.

**Table 2 ijerph-18-09365-t002:** Cases of snakebite with reported information on the genus of the snake, by race group, Brazil, 2017.

Race	*Bothrops*		*Crotalus*		*Micrurus*		*Lachesis*		Non-Venomous		Ignored		Total	
	N	%	N	%	N	%	N	%	N	%	N	%	N	%
White	2751	74.3	355	9.6	22	0.6	17	0.5	233	6.3	323	8.7	3701	100
Black	938	73.9	125	9.8	9	0.7	21	1.6	47	3.7	130	10.2	1270	100
Asian	99	78.0	9	7.1	1	0.8	1	0.8	6	4.7	11	8.7	127	100
Mixed	7474	76.1	698	7.1	74	0.8	254	2.3	413	4.2	910	9.3	9823	100
Indigenous	779	79.7	53	5.4	5	0.5	51	5.2	31	3.2	58	5.9	977	100
Total	12,041	75.7	1240	7.8	111	0.7	344	2.2	730	4.6	1432	9.1	15,898	100

**Table 3 ijerph-18-09365-t003:** Number of snakebites by time between accident and health care, by race, rural population, Brazil, 2017.

Variable	Total (%)	Indigenous (%)	White (%)
**Time between** (*N* = 15,525)			
0–1 h	4113 (26.49)	129 (13.54)	1491 (40.92)
1–3 h	5384 (34.68)	222 (23.29)	1250 (34.30)
3–6 h	2949 (19.00)	201 (21.01)	492 (13.50)
6–12 h	1315 (8.47)	131 (13.75)	152 (4.17)
12–24 h	713 (4.59)	128 (13.43)	88 (2.41)
24 h+	591 (3.81)	108 (11.33)	76 (2.09)
Ignored	460 (2.96)	24 (3.57)	95 (2.61)
**Case classification** (*N* = 15,466)			
Mild	8171 (52.83)	470 (49.01)	1928 (53.16)
Moderate	5745 (37.15)	368 (38.37)	1264 (34.85)
Severe	1195 (7.73)	104 (10.84)	334 (9.21)
Ignored	355 (2.30)	17 (1.77)	101 (2.78)
**Received antivenom** (*N* = 15,898)			
Yes	13,070 (82.21)	830 (84.95)	2949 (79.68)
No	2290 (14.40)	132 (13.51)	614 (16.59)
Ignored	538 (3.38)	15 (1.54)	138 (3.73)

**Table 4 ijerph-18-09365-t004:** Evolution of snakebite cases (death by snakebite, local complication, systemic complication), by race, rural population, Brazil, 2017.

Variable	Total (%)	Indigenous (%)	White (%)
**Evolution** (*N* = 14,260)			
Cure	13,583 (95.25)	849 (97.03)	3205 (94.24)
Death by snakebite	52 (0.36)	6 (0.69)	6 (0.18)
Death by other causes	9 (4.32)	1 (0.11)	3 (0.09)
Ignored	616 (4.32)	19 (2.17)	187 (5.50)
**Local complications ^a^** (*N* = 14,631)			
Yes	569 (3.89)	53 (5.90)	113 (3.25)
No	13,413 (91.68)	809 (90.09)	3193 (91.73)
Ignored	649 (4.44)	36 (4.45)	175 (5.03)
**Systemic complications ^b^** (*N* = 14,397)			
Yes	173 (1.20)	14 (1.60)	46 (1.33)
No	13,536 (94.02)	824 (93.96)	3225 (93.48)
Ignored	688 (4.78)	39 (4.45)	179 (5.19)

Legend: ^a^ Local complication (*n* = 569): secondary infectious (*n* = 418, 73.46%); necrose (*n* = 90, 15.82%); compartmental syndrome (*n* = 65, 11.42%); functional deficit (*n* = 71, 12.48%); amputation (*n* = 7, 1.23%). ^b^ Systemic complication (*n* = 173): renal failure (*n* = 137, 79.19%); respiratory failure/acute pulmonary edema (*n* = 56, 32.37%); septicemia (*n* = 16, 9.25%); shock (*n* = 37, 21.39%).

**Table 5 ijerph-18-09365-t005:** Number of bad outcomes by snakebites (death by snakebite and/or local complication and/or systemic complication) and covariables, and results of univariate and multivariate analysis, rural population, Brazil, 2017.

Variable (*N* = 13,710)	Frequency		Rate of Bad Outcome	Odds Ratio Univariate Model		Odds Ratio Multivariate Model	
	N	%		OR	IC95%	OR	IC95%
**Race**							
White	3267	23.8%	4.2%	1		1	
Black/Mixed	9582	69.9%	4.7%	1.13	0.93–1.38	1.13	0.92–1.39
**Indigenous**	861	6.3%	6.6%	1.63	1.19–2.24	1.18	0.83–1.67
**Location on the body of the bite**							
Hand and arm	2957	21.6%	4.0%	1		1	
Feet and leg	10,572	77.1%	4.9%	1.23	1.00–1.50	1.19	0.96–1.47
Head and torso	181	1.3%	3.3%	0.82	0.35–1.88	0.75	0.32–1.76
**Age**							
0 to 19	3205	23.4%	5.2%	1		1	
20 to 49	6816	49.7%	4.1%	0.79	0.65–0.96	0.81	0.66–1.00
50 to 64	2582	18.8%	4.9%	0.94	0.74–1.19	0.95	0.74–1.22
64 up	1107	8.1%	6.1%	1.20	0.90–1.60	1.16	0.85–1.57
**Sex**							
Male	10,480	76.4%	4.5%	1		1	
Female	3230	23.6%	5.2%	1.15	0.96–1.38	1.19	0.98–1.43
**Received antivenom**							
Yes	11,739	85.6%	5.1%	1		1	
No	1971	14.4%	1.9%	0.36	0.26–0.51	0.71	0.49–1.00
**Case classification**							
Mild	7341	53.5%	1.7%	1		1	
Moderate	5268	38.4%	5.9%	3.60	2.91–4.44	3.29	2.65–4.01
Severe	1101	8.0%	18.9%	13.44	10.6–16.9	11.8	9.29–15.07
**Time between accident and health care**							
0–3 h	8559	62.4%	3.6%	1		1	
3–6 h	2729	19.9%	5.5%	1.59	1.30–1.94	1.38	1.12–1.70
6–24 h	1888	13.8%	6.5%	1.87	1.51–2.33	1.48	1.18–1.86
24 h+	534	3.9%	12.2%	3.76	2.83–4.99	2.99	2.21–4.06

**Table 6 ijerph-18-09365-t006:** Number of snakebites with information of all covariables, number of bad outcomes of the accident and percentages, rural population, Brazil, 2017.

Sates	N Bites	N Bad Outcome	%
Rondônia	256	29	11.3
Acre	226	9	4
Amazonas	1012	68	6.7
Roraima	260	13	5
Pará	3022	148	4.9
Amapá	205	22	10.7
Tocantins	252	9	3.6
North	5233	298	5.7
Maranhão	670	41	6.1
Piauí	144	6	4.2
Ceará	432	8	1.9
Rio G. Norte	133	3	2.3
Paraíba	177	3	1.7
Pernambuco	272	8	2.9
Alagoas	100	9	9
Sergipe	34	1	2.9
Bahia	1248	41	3.3
Northeast	3210	120	5.7
Minas Gerais	1536	55	3.6
Espírito Santo	464	4	0.9
Rio de Janeiro	160	6	3.8
São Paulo	654	29	4.4
Southeast	2814	94	3.3
Paraná	458	18	3.9
Santa Catarina	244	10	4.1
Rio G. Sul	486	28	5.8
South	1188	56	4.7
Mato G. Sul	268	10	3.7
Mato Grosso	555	36	6.5
Goiás	430	27	6.3
D. Federal	12	1	8.3
Central- West	1265	74	5.8
Brazil	13,710	642	4.7

## Data Availability

Aggregated data by municipality can be downloaded from: Ministério da Saúde do Brasil. Sistema de informação de agravos de notificação (SINAN). Available online: http://www.portalsinan.saude.gov.br/doencas-e-agravos (accessed on 18 August 2019). Individual cases database can be requested from the Brazilian Government from: Governo Federal do Brasil. Acesso a informação. Available online: https://www.gov.br/acessoainformacao/pt-br (accessed on 19 March 2019).

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
