# Peer review of "Snakebites in Rural Areas of Brazil by Race: Indigenous the Most Exposed Group"

_ijerph, 2021, doi:10.3390/ijerph18179365_

Round 1

Reviewer 1 Report

A comprehensive synopsis of Brazil's rural health service in relation to snakebites. Overall the paper is well written, interesting and this work will add to the paucity of data on snakebites in South America. 

Attached are comments for authors

Reviewer 2 Report

The manuscript entitled “Snakebites in rural areas of Brazil by race: Indigenous the most exposed group” presents a retrospective epidemiology study of snakebite cases in Brazil to identify which racial group is most exposed and factors that contribute to bad outcomes. The manuscript is well-written and represents a relevant topic to Brazil’s Public Health, especially due the fact that the distribution of antivenoms is not well-balanced, and most affected regions frequently do not have access to it.

Figure 1 shows the variability of distribution of snakes in Brazil, and that North and Northeast regions have the coexistence of genera, and this is an important point that was not explored in the manuscript. Besides the fact that the treatment is time-dependent for a good outcome, another relevant aspect that may have influence in mortality could be related to species that caused the accident. Moreover, there are at least 3 types of antivenoms in Brazil (antibothropic and antilachetic, anticrotalic and antielapidic) and a crucial step is to identify the snake species to administrate the correct antivenom. Although most cases are usually related to Bothrops genera, I’m curious to known if authors checked about snake species that caused accidents in indigenous. Was the species identified in most cases, or is it unknown (blank/ignored)? The identtification of snake species is comparable with the rest of races in rural areas? I believe this could be discuss in a brief paragraph.

Minor points

1) Line 155 “In case of an accident with a poisonous animal”

Please avoid the word poisonous when referring to snakebites. It should replace by venomous, such as the rest of the text.

2) Line 218-219 “calculated for the most exposed and lees s exposed racial groups.”

Less exposed racial groups?

3) Line 273 -274 “ this probability is even higher (8.9), whereas it is lower in the Southeast (1.8) and South (2.5)”

My suggestion is to include “x” or “times” after the numbers to be more clear – i.e “ this probability is even higher (8.9x), whereas it is lower in the Southeast (1.8x) and South (2.5x) “

4) Line 278 “Legend: NA: Not applicable in the states with less of 1,000 population in these race group. .”

Delete the extra period.

5)Table 4 needs a revision in its format -  In "Frequenty" column,  N and % are very close, and some numbers are confusing (i.e  Local of the body of the bite is shown 10,57277.1% )
